# Evidence from the Synthetic Laboratory: Language Models as Auction Participants

## Abstract

This paper investigates the behavior of simulated AI agents (large language models, or LLMs) in auctions, validating a novel synthetic data-generating process to help discipline the study and design of auctions. We begin by benchmarking these LLM agents against established experimental results that study agreement or departure between realized economic behavior and predictions from theory; i.e., revenue equivalence between first-price and second-price auctions and improved play in obviously strategy-proof auctions. We find that when LLM-based agents diverge from the predictions of theory, they do so in a way that agrees with behavioral traits observed in the existing experimental economics literature (e.g., risk aversion, and weak play in 'complicated' auctions). Our results also suggest that LLMs are bad at playing auctions 'out of the box' but can improve their play when given the opportunity to learn. This learning is robust to various prompt specifications and holds across a variety of settings. We run 2,000+ auctions for less than $250 with GPT-4o and GPT-4, and develop a framework flexible enough to run auction experiments with any LLM model and a wide range of auction design specifications.

## 1 Introduction

The design of economic mechanisms, especially auctions, has benefited enormously from a rich interplay between empirics and theory. One recent example is the development of *obviously strategy-proof mechanisms* (hereafter, OSP) (Li, 2017). The technical refinement from strategy-proof to obviously strategy-proof was inspired by an empirical puzzle: despite it being well-known that the open-ascending clock and second-price sealed-bid auctions were strategically equivalent, experiments since the 80s suggested that people were much better at playing the open-ascending clock auction than the sealed-bid auction (Kagel et al., 1987). Motivated by empirical evidence, OSP provided one articulation for why the clock format might be better than the sealed-bid format and has since inspired a flourishing of work in mechanism design under behavioral constraints (E.g., Nagel & Saitto (2023); Pycia & Troyan (2023); Gonczarowski et al. (2023)). The story echoes a well-understood but worth emphasizing point: empirical work is vital to the development of new theory.

Unfortunately, empirical evidence is quite expensive to generate. Li (2017)'s OSP experiments alone, with 404 participants, cost over $15,000.[1] Given this, the rise of LLMs raises the exciting new question of whether there exist cheaper data-generating processes that can (partially) substitute for human data for the purpose of studying human behavior, whether in economic systems or otherwise (Bubeck et al. (2023); Horton (2023); Manning et al. (2024)). The present work examines this question for auctions: how well do LLMs substitute for humans in auctions, especially when behavioral traits like risk-aversion or bounded rationality matter? And, more generally: how can we use LLMs to improve the design of economic mechanisms?

After a brief literature review, we'll begin with a description of the simulation procedure to fix ideas. The bulk of the work proceeds in two parts from there. In Section 3.1, we study classic, textbook results using minimally tailored LLM agents (Krishna, 2009). Namely, results such as revenue

---

[1] Li mentioned that for each participant, he would pay them $20 for participation and an additional money prize they won during the game. In total, he paid on average $37.47 for each participant.

equivalence (and associated predictions about strategic play) under the first-price sealed bid (FPSB) and second-price sealed bid (SPSB) auctions in independent, private value (IPV) settings. When there are departures from theory, we are interested in whether the departures agree with existing experimental results (Kagel & Roth, 2020). We find that, with LLM agents, bids in the SPSB auction are higher than bids under the FPSB auction (as would be expected), but that there is a smaller separation between the two than would be predicted by theory. This is primarily due to bids under the FPSB auction being higher than the prediction coming from a risk-neutral Bayes-Nash equilibrium. One possible explanation is that LLMs play according to some level of risk aversion: this is consistent with Cox et al. (1988)'s survey of over 1,500 IPV auction experiments, which finds that revenue under the FPSB auction is higher than under the SPSB auction due to risk aversion. However, we also see LLMs in the SPSB auction tend to submit bids lower than their value to an extent that is not echoed in the experimental literature.

For completeness, we also conduct semantic analysis in Appendix A.8.2 to investigate both of these results further – having endowed LLM agents with chain-of-thought reasoning, we can parse this reasoning along three dimensions: understanding (how well the LLM seems to understand the auction), aggression (how aggressive the LLM's plans seem to be), and interdependency of strategies (how reactive the LLM's strategy is to actions other LLMs take). We find that most of the divergence from the current experimental literature can be attributed to low 'aggression' (i.e., that LLMs demonstrate even more risk-aversion than laboratory participants), and further corroborate this with an intervention that prompts LLMs to be less risk averse.

Having investigated differential behavioral responses between auction formats, we next seek to understand whether LLMs exhibit behavior similar to humans in the face of strategic complexity. In Section 3.2, we make the auction format itself 'easier' to play—switching from sealed-bid formats to clock formats—and show that LLMs play closer to theoretical predictions as a result. Theoretical and empirical benchmarks suggest clock auctions induce more rational play in humans, and our experiments test whether LLM agents also play clock auctions more rationally than sealed-bid auctions. Our experimental design is inspired by theory developed in Li (2017), and we find that in affiliated private value (APV) settings, LLM behavior agrees with human behavior—LLMs play more closely to the predictions of economic theory for ascending clock auctions. For robustness, we also run these experiments in an IPV setting and obtain the same results. In both settings, we compare the ascending second-price clock auction with a 'blind' ascending clock auction (where bidders do not know when other players drop out and the auction stops at the final drop out) and the SPSB auction.

As with other empirical work, the "interface" that an experimenter uses with subjects can greatly impact results. The Appendix reports prompts and robustness checks amongst different prompting schemes that we have tried with LLM agents. In particular, for each experiment, we used default 'technical' prompts that closely followed that of the Appendix A.1 script from Li (2017) and also report results from non-technical 'humanistic' prompts whose results were very similar. We also find interesting evidence that 'goal' prompting (e.g., reminding the LLM that the goal is to maximize profit) leads to bidding that more closely follows rational economic theory, leading to higher allocative efficiency [2].

To obtain the data for these empirical results, we have developed a code repository [3] to systematically run experiments with any number of bidders and any set of prompts. In particular, our repository is flexible enough that it can be used to generate synthetic data for almost any describable auction format and for auctions with single or multiple goods. For the experiments herein, we ran more than $2,000$ auctions with more than $5,000$ GPT-4 and GPT-4o agent participants for a cost totaling less than $250 in API calls. In contrast, the largest survey of auction experiments to date comes from Cox et al. (1988) of $1,500+$ auctions, with total costs likely considerably higher. [4].

---

[2]See also Manning et al. (2024) for related discussion. 'Goal' prompting seems to give LLM agents a clear objective to anchor on. Without such anchoring, LLM agents are unwilling to profit maximize.

[3]The code repository will be made public once accepted.

[4]Modestly considering an average cost of $20 per person and 3 people per laboratory auction, the cost of all the auctions in Cox's survey can be estimated to have been at least $90,000.

## 1.1 RELATED WORK

**Auctions:** There's a vast quantity of theoretical and experimental literature on auctions. While Krishna (2009)'s textbook and Kagel & Roth (2020)'s handbook provide invaluable general resources, we will stay focused only on the citations relevant to the results in this paper.

Starting with the IPV case, the theoretical benchmark of revenue equivalence in the risk-neutral case is exposited in the seminal Myerson (1981). The experimental evidence for departures due to risk-aversion in the FPSB auction is thoroughly documented in Coppinger et al. (1980) and Cox et al. (1988)'s survey. The experimental evidence for the common error of bidding above one's value in the SPSB auction is documented in Kagel & Levin (1993).

Recent work on obvious strategy-proofness began with Li (2017), who demonstrates empirically that human subjects tend to be more truthful in second price sealed bid auctions than ascending clock auctions in the APV setting, even though the two auctions are strategically equivalent and provides the new theoretical framework of OSP to explain the results. To better understand our simulations, we also consider the experimental evidence presented by Breitmoser & Schweighofer-Kodritsch (2022), who investigate intermediate auction formats that decompose the behavioral effects in Li (2017).

**LLMs as simulated agents:** Recent work has shown that LLMs, having been trained on an enormous corpus of human-generated data, are able to generate language and reasoning patterns (Achiam et al., 2023; Bubeck et al., 2023). Yet, they are far from perfect and show limited planning abilities and various cognitive biases (Wan et al., 2023). There is also a growing literature on using these human-like AI models as simulated agents in economics and social science studies (Aher et al., 2023; Park et al., 2023; Brand et al., 2023). Horton (2023), in particular, replicates four classical behavioral economics experiments by endowing a single LLM agent with different personas and querying it about its decisions and preferences.

**LLMs in auctions:** There are a few works on systematically using LLMs as simulated agents in auction experiments. Fish et al. (2024) study the collusion behaviors in first-price sealed-bid auction of two LLM agents under the context of LLMs as a price setter for companies. Chen et al. (2023) study how to make an LLM better at playing auctions than humans in a dynamic game with multi-items. Manning et al. (2024) report results from a variant of open-ascending clock auction with three LLM agents, focusing on deviations from rational economic theory in considering bidders' values and the final clearing price.

## 2 METHODS

### 2.1 LLM AGENT DESIGN

In each experiment, we simulate $n$ (often, 3) LLM agents to play against one another in an auction. Each setting is repeated 15 times with values drawn randomly each time. The specification, seen below, is designed to closely follow actual multi-round human laboratory auction experiments run in the literature. In all games, LLM agents bid for a prize (specified as 'prize' in the technical prompts). If an agent wins the prize, they earn an amount equal to the value of the prize, minus their payment in the auction. In all settings, values are drawn from distributions with support over [\$0, \$99]. Bids in these games are in \$1 increments primarily to reduce token usage, but can easily be made more fine.

#### 2.1.1 SIMULATION PROCEDURE

We follow the following simulation procedure for each auction. Full prompt texts are provided in Appendix A.1.

1. LLMs are first briefed on the auction format and then asked to generate a plan for bidding.

2. Upon making a bidding plan, the system endows LLMs with a realized random value.

3. Given their value, LLM agents are then asked to make a bid according to their plan.

4. Given all bids, the system determines allocations and transfers for each LLM agent. LLMs are informed whether they've won or lost the auction as well as their realized profit. LLMs also see the entire distribution of bids (but, of course, not everyone's value).

5. Before entering the next round, LLM agents are also instructed to reflect upon their bidding strategy and result. This information is all saved in the HISTORY variable as one round.

6. One such 'plan-bid-reflect' loop constitutes one round. The system conducts 15 such rounds in one experiment. Before each round, LLM agents are given the entire HISTORY variable up to that point.

Interestingly, LLM agents are much better at playing auctions after explicitly being told their objective is to profit maximize. To set LLMs as profit maximizers, we appended a prompt prefix to the rules of each auction. This approach was inspired by Fish et al. (2024):

```
Your TOP PRIORITY is to place bids which maximize your
profit in the long run. To do this, you should explore
many different bidding strategies, including possibly
risky or aggressive options for data-gathering purposes.
Learn from the history of previous rounds in order to
maximize your total profit. Don't forget the values are
redrawn independently each round.
```

We implemented these prompts using the EDSL framework developed by Horton et al. (2024) for querying LLM agent's decision in the desired format. Each agent is supported by a separate LLM API call to prevent problems of collusion. For every auction in this paper, we used models from OpenAI and set the temperature to $1$.[5]

## 3 BENCHMARKING WITH PREVIOUS AUCTION EXPERIMENTS

### 3.1 FIRST-PRICE VERSUS SECOND-PRICE SEALED BID AUCTION

We begin by examining the First-Price Sealed-Bid (FPSB) and Second-Price Sealed-Bid (SPSB) auctions in an independent private values (IPV) setting, where each bidder knows their own valuation of the auctioned item, but not the valuations of other bidders.

### 3.1.1 SETTING

There are three bidders in each auction, and each bidder $i$'s value is drawn from an independent, uniform distribution $v_i \sim U[0, 99]$. Bidder $i$ submits a sealed-bid $\beta_i(v_i)$, and we write $\beta(v) = (\beta_1(v_1), \beta_2(v_2), \beta_3(v_3))$ for the vector mapping each agent's value to its corresponding bid.

In the FPSB auction, the highest bidder pays her bid and receives the prize (and all other bidders pay 0 and receive no prize). Formally, the payment for an agent is given by: $t_i(\beta(v)) = \mathbb{1}_{i \text{ won the auction}} \cdot \beta_i(v_i)$. In the SPSB auction, the highest bidder pays the second-highest bid and receives the prize (and all other bidders pay 0 and receive no prize). Formally, the payment for an agent is given by: $t_i(\beta(v)) = \mathbb{1}_{i \text{ won the auction}} \cdot \beta^{(2)}(v)$, where $\beta^{(2)}$ represents the second-order statistic or the second-largest bid. Bids are submitted in \$1 increments and ties are resolved randomly.

Notation is summarized as follows:

| Object | Notation |
|--------|----------|
| Values | $v \sim U[0, 99]$ |
| Bid | $\beta(v)$ |
| Allocation | $x(\beta)$ |
| Transfer | $t(\beta)$ |

---

[5]We set a non-zero temperature to induce diversity in plans and reflections. No tuning on temperature or other such parameters was conducted to obtain results.

### 3.1.2 THEORETICAL BENCHMARKS

It is well-known that in the SPSB auction with IPV, bidding one's true value is a dominant strategy equilibrium (Vickrey, 1961). This means that, regardless of what strategies other bidders undertake, it is always in an agent's best interest to bid their true valuation for the item being auctioned. Hence, we say the SPSB is *dominant strategy incentive compatible* (DSIC) and can express the equilibrium bidding strategy in an SPSB auction as:

$$\hat{\beta}^*_{SPSB}(v) = v \tag{1}$$

where $\hat{\beta}^*_{SPSB}(v)$ represents the equilibrium bidding strategy in the SPSB auction. The asterisk denotes that the equilibrium is in dominant strategies.

The First-Price Sealed-Bid (FPSB) auction, on the contrary, does not have an equilibrium in dominant strategies. Instead, it has a Bayes-Nash equilibrium (BNE) bidding strategy when values are uniformly distributed with common support of:

$$\hat{\beta}_{FPSB}(v) = \frac{n-1}{n}v \tag{2}$$

Note that, compared with the dominant strategy in SPSB, this equilibrium holds only under certain assumptions on the distributional structure of values (i.e., uniform distribution of valuations) and the structure of agents' von Neumann-Morgenstern preferences (i.e., risk-neutrality). We fix the value distribution as $v_i \sim U[0, 99]$ for all LLM agents in our simulations. Given we're interested in benchmarking the risk preferences of pre-trained LLMs, we do not attempt to condition agents' risk preferences via prompting.

### 3.1.3 EMPIRICAL BENCHMARKS

Results from laboratory experiments offer additional empirical benchmarks to support and refine theoretical predictions.

In both the First-Price Sealed-Bid (FPSB) and Second-Price Sealed-Bid (SPSB) auctions (and indeed, almost all auctions), there is robust experimental evidence of bids being strictly monotone in one's value. This means that higher valuations consistently lead to higher bids, aligning with theoretical predictions. However, the specific bidding patterns observed in experiments often deviate from theoretical equilibria in interesting ways.

**Empirical evidence for FPSB:** In the case of FPSB auctions, experimental evidence consistently shows bids above the risk-neutral BNE prediction. Cox et al. (1988) found that participants in FPSB auctions tend to bid more aggressively than the theory predicts for risk-neutral bidders. In an FPSB auction, bidding higher increases the probability of winning but decreases the potential profit if one does win. Risk-averse bidders may be willing to accept lower potential profits in exchange for a higher chance of winning, leading to bids above the risk-neutral BNE

**Empirical evidence for SPSB:** Interestingly, experimental data for SPSB auctions reveals a different pattern of deviation from theoretical predictions. Participants often bid higher than than their true valuations despite the dominant strategy of truthful bidding in SPSB auctions. Kagel and Levin's seminal study (Kagel et al. (1987); Kagel & Levin (1993)) provides detailed insights into this behavior:

1. Overbidding: A significant proportion of participants (typically around 60-70%) in SPSB auctions submitted bids above their true values. This contrasts sharply with the theoretical prediction of truthful bidding.
2. Learning Effects: While overbidding persisted throughout the experiments, its frequency and magnitude tended to decrease over time as participants gained experience with the auction format.

These findings challenge the straightforward theoretical predictions for SPSB auctions and highlight the complexity of human behavior in auction settings. Several explanations have been proposed for this overbidding phenomenon, including: misconceptions about the second-price rule (Kagel et al. (1987)), strategic uncertainty, and beliefs about others' irrationality (Crawford & Iriberri (2007)).

### 3.1.4 SIMULATION EVIDENCE

We performed 5 parallel simulations with 15-rounds for each of the SPSB and FPSB auctions with 3 bidders. The SPSB bidding data is plotted in the left panel of Figure 1, and the FPSB bidding data is plotted on the right. The X-axis represents LLM agent's assigned values for the good. The Y-axis is the bid placed by the LLM-agent at the given value. The grey triangles are the actual LLM experiment data. The black dashed line is the Loess-smoothed curve for all the experimental data. The 45-degree line indicates the scenario where the LLM agents' values equal their bids while the solid red line is the theoretically optimal bidding strategy.

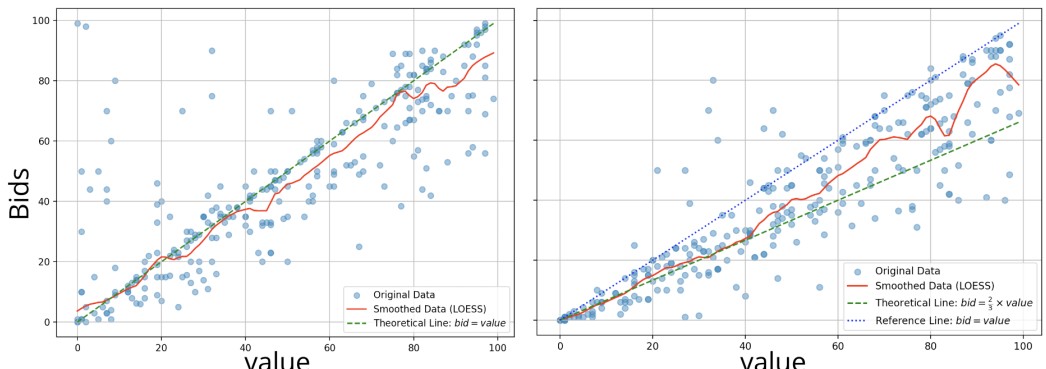

Figure 1: **Comparison of SPSB and FPSB under IPV setting.** The theoretically predicted bid, given that bidders' values are independently drawn from a uniform distribution of [0, 99], is marked in red. The experimental data points are represented by grey triangles. The 45-degree line indicates the scenario where the LLM agents' values equal their bids. The dashed black line represents the LOESS-smoothed data. Left: In the SPSB experiment results, the bids are shading down compared to the dominant strategy. Right: In the FPSB experiment results, the bids are shaded up relative to the Bayes-Nash prediction.

Both sets of data demonstrate evidence of monotone bidding – the most stable hallmark of the empirical literature on auctions. For the FPSB results, the Loess-smoothed data curve is significantly higher than the Bayes-Nash equilibrium predicts. However, there is significant mass above and below the risk-neutral Bayes-Nash prediction, which closely matches the existing empirical evidence in first-price auctions (Cox et al., 1988).

For the SPSB results, the Loess-smoothed data curve approximately aligns with the dominant strategy prediction of bidding one's value. While the majority of bids are close to the values, noticeably, for SPSB, there are also many high bids corresponding to low values. Uniformly low bids are also prevalent across the value range. These failures of monotonicity are not prevalent in existing empirical literature, and primarily represent LLM's executing plans which ignore their value. LLMs also rarely bid above their value, whereas existing empirical literature finds the opposite – usually, people find the inefficiency of bidding above one's value to be a subtle point in the SPSB auction (Kagel et al., 1987).

For cross-format comparisons, we plot the bidding behavior of LLM agents in Appendix Fig. 11.

Finally, to further quantify the difference between the two auction formats, we conducted an independent samples t-test, obtaining a T-statistic of $-3.22$ with an associated p-value $0.0013 < 0.01$, suggesting a statistically significant difference in bidding behavior between the FPSB and SPSB auctions.

## 3.2 OBVIOUS STRATEGYPROOFNESS

After examining sealed-bid auction formats, we now turn our attention to dynamic auction formats, specifically clock auctions, and compare them to sealed-bid formats. Clock auctions are dynamic auction formats where the price increases incrementally (or decreases in reverse auctions), and bidders decide whether to stay in or drop out at each price point. These auctions are strategy-proof

mechanisms since a player does not need to form beliefs about other players' values and types. She can simply report her value truthfully (bid her value) because she can do no better regardless of what the other players bid or what their bidding strategies are. That is, truthful reporting is a weakly dominant strategy.

In practice, many people do not report their types truthfully in strategy-proof mechanisms. Li (2017) introduces obviously strategyproofness as a refinement of strategyproofness to explain why certain interactive mechanisms might be easier to recognize as strategyproof.

### 3.2.1 Setting

There are 3 bidders in each auction but now bidders draw affiliated private values of the form $v = c + p$. The common component is drawn uniformly $c \sim U[0, 19]$ and the private component is drawn uniformly $p \sim U[0, 20]$. Winners of the auction receive their own value of the prize $v$ when they win, so the 'common' and 'private' components only serve to make values correlated (even if draws are independent). The ascending clock auction (called AC below) is the classic English auction. The blind ascending clock auction (called AC-B below) is the English auction with the addition of not being told when other bidders leave. The SPSB auction was defined above.

All three of the auction formats in this case are strategically equivalent to second-price auctions, so the affiliation in values is, in a sense, a red herring – for all three auctions it is still dominant strategy to bid one's value. The two clock auctions are obviously strategyproof, though the AC-B auction still provides bidders with 'less' information than the AC auction. The affiliation hence serves only to complicate the auction for bidders who don't appreciate that the dominant strategy is to bid one's value.

### 3.2.2 Theoretical benchmarks

The ascending clock (AC) auction is strategically equivalent to the second-price sealed-bid (SPSB) auction, so it's still DSIC to bid one's value:

$$\hat{\beta}^*(v) = v \tag{3}$$

where $\hat{\beta}^*(v)$ is the optimal bid for a bidder with value $v$. This theoretical benchmark suggests that under both formats, bidders should reveal their true valuation, and we say that both auctions are strategy-proof (SP).

The refinement of OSP, introduced by Li (2017), provides a formal framework for analyzing the "cognitive simplicity" of mechanisms. Consider a strategy $S_i$ for player $i$. We say that $S_i$ obviously dominates another strategy $S_i'$ if, for any deviation from $S_i$ to $S_i'$, the best possible outcome from $S_i'$ is no better than the worst possible outcome from $S_i$. In the ascending clock auction, the price rises incrementally so that bidders only need to consider their decision to stay in at the given price. Hence, we say that the ascending clock auction is also obviously strategy-proof.

Intuitively, a mechanism is obviously strategy-proof (OSP) if players can recognize their optimal strategy without having to reason through hypothetical counterfactual scenarios.

Formally, for all histories $h$ where both $S_i$ and $S_i'$ are consistent with $h$:

$$\min_{z \in Z(S_i, h)} u_i(z) \geq \max_{z' \in Z(S_i', h)} u_i(z') \tag{4}$$

Where $Z(S_i, h)$ is the set of terminal histories that can result from playing $S_i$ after history $h$, and $u_i(z)$ is player $i$'s utility for terminal history $z$.

### 3.2.3 Empirical benchmarks

Li (2017)'s experiment delivers results supporting the theoretical framework of obvious strategyproofness – even though the AC and SPSB auctions are strategically equivalent, human subjects tend to be more truthful under the AC auction (which is OSP) than under the SPSB auction.

Additional empirical results by Breitmoser & Schweighofer-Kodritsch (2022) show that even OSP itself might not be sufficient in capturing the rich complexities of human behavior – human subjects

are less truthful under AC-B than they are under AC (though still more truthful than the SPSB), even though both AC and AC-B are OSP. They show that alternative framing of the same auction format can improve the rate of straightforward behavior.

The empirical paper by Gonczarowski et al. (2023) takes a deeper look at the framing of SPSB. They experimented with describing/framing a sealed-bid auction with Nash deviation description and observed significantly more truthful bidding than traditional descriptions.

### 3.2.4 SIMULATION EVIDENCE

Figure 2 summarizes our findings in the APV setting. In particular, play improves dramatically as the strategic complexity of the environment decreases. In the SPSB, agents often misbid, while they play comparatively much better in the AC-B (AC auction but without information on who else is still in the auction) and the AC auction.

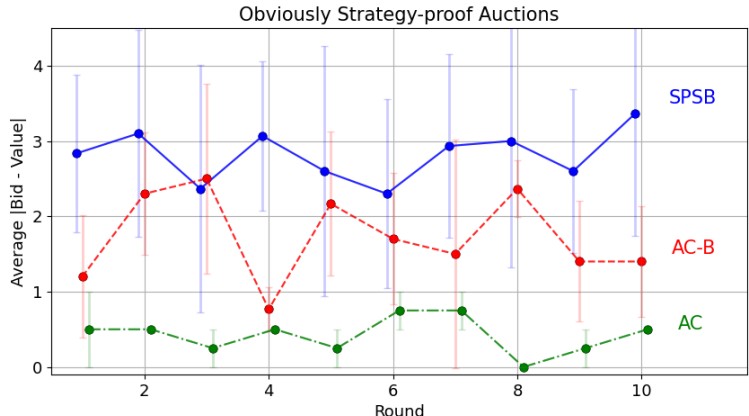

Figure 2: **Comparison of three strategically equivalent auctions.** Ascending-clock (AC) and its variant without dropping-out information (AC-B) are obviously strategy-proof while second-price sealed-bid (SPSB) is not. Here, the green dot-dash line plots the mean absolute deviation between bids and values in the AC; the red dash line for the AC-B; and the blue solid line for the SPSB.

Two-sample t-tests of the mean absolute deviations between bids and values show significant differences across all comparisons of the strategy-proof mechanisms (see Table 1). AC exhibited significantly smaller deviations compared to both AC-B (t = -4.125, p < 0.001) and 2P (t = -5.413, p < 0.001). Furthermore, the AC-B mechanism shows significantly smaller deviations than the 2P auction (t = -5.006, p < 0.001).

| Comparison | t-statistic | p-value |
|---|---|---|
| AC v.s. AC-B | -4.125 | 6.737e-05 |
| AC v.s. 2P | -5.413 | 2.101e-07 |
| AC-B v.s. 2P | -5.006 | 1.043e-06 |

Table 1: T-Test Results for Strategyproof mechanisms

Interestingly, we see little evidence of learning over time. In the experiments in Li (2017) and Breitmoser & Schweighofer-Kodritsch (2022), human subjects improve their understanding of the mechanisms by bidding closer to their true value over time. In the results presented in Figure 2, this effect isn't as pronounced. We conjecture that clever prompting strategies, leading to a better and more natural way of communicating the history of past rounds, might lead to more human-like behavior exhibiting learning over rounds. We further plotted the bids versus values for all the rounds in Fig 3.

In the AC auction, all players dropped out at the dominant strategy price within one round of bidding. The bids in AC-B auction are close to the optimal line, but demonstrate a significant pattern of early

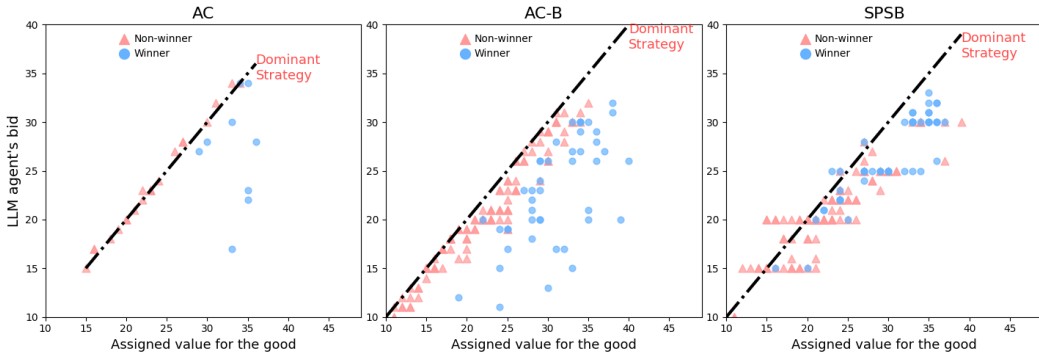

The x-axis is the assigned value for the good and Y-axis is LLM agent's bid. Red triangle stands for the bids of non-winner in the auction while the blue circle represents the ones of the winners.

Figure 3: **Bid versus value scatter plot for different auction formats.**

dropouts. In the SPSB auction, almost all the bids deviate from the dominant strategy, diverging both higher and lower than the optimal price.

We conduct further robustness checks in Appendix A.8, including conducting 'interviews' with LLM agents to ask them to explain their bidding behavior and counterfactual experiments.

## 4 CONCLUSION

This paper reports the results of more than 2,000 auction experiments with LLM agents. In particular, we find behavior that conforms with important experimental results (i.e., evidence of risk-averse bidding and evidence that clock auctions are 'easier' to play). Though the results are encouraging, we see this work as preliminary, primarily putting forward a framework on how to think about LLM experimental agents as proxy for human agents. In particular, the design space for prompting is large, and we hope that interested readers will use our code to run simulations testing their own prompt variations.

In addition, while this paper focuses on auction theory, future work may use LLM sandboxes to test other kinds of economic mechanisms (e.g., voting, matching, contracts, etc.). As techniques are developed to validate LLM models as proxies for human behavior, they can be used to obtain what would otherwise be prohibitively expensive evidence. As a provocative example, while ethical and financial constraints make it impossible to run voting experiments at the scale of nations, it may be possible to run such experiments with LLM agents.

We're particularly interested in the use of such techniques to generate synthetic data which is useful for informing novel economic design. In particular, some auction formats, such as combinatorial auctions, are complex and can be particularly difficult to run frequently and at scale in traditional lab experiments. Augmenting these traditional lab experiments with LLM experiments, when correctly validated, may open up wide new avenues to better understand the design tradeoffs in these kinds of complex and often high-stakes environments.

## ACKNOWLEDGMENTS

We thank EDSL and Robin Horton for technical support in deploying the simulation environment.

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

## A  APPENDIX

### A.1  SIMULATION PROCESS OF IPV AUCTIONS

During the auction, LLM agents will be first informed about the scenario. After explaining to them the rules and incentives of the aucion, they will be asked to write down a plan for the bidding.

```
You are Bidder Andy.
You are bidding with Bidder Betty, Bidder Charles.

{RULE EXPLANATION} + {INSTRUCTIONS} + {PERSONA}

Write your plans for what bidding strategies to test next.
Be detailed and precise but keep things succinct
and don't repeat yourself.
Your plan should be within 100 words.
```

The following is the output from Bidder Andy for the first round in a FPSB auction:

> *"I plan to initially explore a strategy where I bid half of my value, then varying my bids between 30-70% of my value in subsequent rounds. Sequentially, I will introduce risk by bidding within +/- 5% of the highest bid from previous round, gathering data about other bidder's tendencies. Lastly, based on gathered data, I'll deploy a more aggressive high bid if losses amass, or low-risk bid if leading, to secure profit. Learning from each round is vital to adapt and refine strategy for profit maximization."*

Once they make the bidding plan, the system will inform each LLM agent about their value towards the item in this round and ask them to make a bid according to their plan.

```
You are Bidder Andy.
You are bidding with Bidder Betty, Bidder Charles.

{RULE EXPLANATION} + {INSTRUCTIONS} + {PERSONA}

Your value is 73. Your plan is {PLAN}
FOLLOW YOUR PLAN.
```

```
How much would you like to bid?
Give your response with a single number and no other texts,
e.g. 1, 44
```

In this example, Bidder Andy valued the item at $ 73. And according to his plan, he only wanted to bid 30-70% of his value. So, he bid *"36.5"* in this round.

After receiving the bids from all the LLM bidders the system determines the winner and the profits for each bidders. All the bids and the winner's profit are returned back to the bidders along with whether they win the auction in this round. The part of information is included in the HISTORY variable. Before entering the next round, the bidders are also asked to do a reflection upon their bidding strategy and the result.

```
You are Bidder Andy.
You are bidding with Bidder Betty, Bidder Charles.

{RULE EXPLANATION} + {INSTRUCTION} + {PERSONA}

The previous round history is: {HISTORY}

Do a counterfactual analysis of the last round.
REMEMBER that your goal is to win the bid and make higher
profits. REMEMBER YOUR PAYMENT IS YOUR BID IF YOU WIN.
Let's think step by step.
Start your reflection with
'If I bid down by .., I could...
If I bid up by ..., I could...'
LIMIT your OUTPUT within 100 words.
```

An example of Bidder Andy's reflection is provided below. Since he didn't win the auction and incurred a zero profit, his reflection was bidding up closer to his true value would help him win the game and secure a net profit.

> *"If I bid down by a few dollars, say around $30, I likely still wouldn't have won given the winning bid was $57. I would've merely saved my bid money. If I bid up by $21, matching the winning bid of $57, I may or may not have won due to the random resolution of ties. However, if I'd won, my profit would have been $16, realizing the difference between my value of $73 and my raised bid. So, bidding aggressively could potentially increase my profits in such scenarios. "*

In the following new round, all the bidders were asked to update their plans according to the history of the last round and their reflections.

```
You are Bidder Andy.
You are bidding with Bidder Betty, Bidder Charles.

{RULE EXPLANATION} + {INSTRUCTION} + {PERSONA}

The previous round history is: {HISTORY}
After careful reflection on previous bidding,
your analysis for last round is {REFLECTION}.

Learn from your previous rounds,
Let's think step by step to make sure we make a good choice.
Write your plans for what bidding strategies to test next.
Be detailed and precise but keep things succinct and don't
repeat yourself. LIMIT your plan to 50 words.
```

Here, we showed the updated plan for the second round of the same Bidder Andy. After learning from the lessons of bidding only 50% of the value and losing the game, this time the agent update his plan to the following:

> *"I plan to test higher bidding strategies, around two-thirds of the prize value to foster winning chances. Additionally, if my prize value is extremely high, I will bid aggressively for data-gathering."*

This plan-bid-reflection loop iterates until the end of the auction. In our setting, there are 15 rounds of this 'plan-bid-reflection loop' in each of the first-price and second-price auctions.

## A.2 SIMULATION PROCESS IN OSP AUCTIONS

In the OSP setting, researchers usually considered the simplicity of mechanism and people's limited cognitive ability when playing the game Li (2017). Therefore, we removed the plan and reflection part during the bidding to simulate cognitively more limited auction participants Raman et al. (2024). Also, we switched to GPT-4o for building the agents since running the clock auction costs 100 times of inferences more than sealed-bid auction. In the following, we shows the whole simulation process of AC, AC-B and SP.

In the first round of ascending clock auction, we will again inform the rules of auction, similar to the one in session 3.1.

```
You are Bidder Andy.
You are bidding with Bidder Betty, Bidder Charles.

{RULE EXPLANATION} + {INSTRUCTIONS} + {PERSONA}

Your value towards to the prize is 26 in this round.
The current price is 10.
Do you want to stay in the bidding?
```

If they didn't decide to drop out in the first round, they would be shown the previous bidding history for the next clock cycle. The history includes the price and how many people had dropped out.

```
You are Bidder Andy.
You are bidding with Bidder Betty, Bidder Charles.

{RULE EXPLANATION} + {INSTRUCTIONS} + {PERSONA}

Your value towards to the prize is 26 in this round.
The previous biddings are:
['In clock round 1, the price was 10,
no players dropped out'].
The current price is 11.
Do you want to stay in the bidding?
```

In AC-B, we don't show the previous biddings and in each clock cycle, we directly query LLM agents' decision.

```
You are Bidder Andy.
You are bidding with Bidder Betty, Bidder Charles.

{RULE EXPLANATION} + {INSTRUCTION} + {PERSONA}

Your value towards to the prize is 26 in this round.
The current price is 11.
Do you want to stay in the bidding?
```

To match the decision process of Ascending clock, for the second-price sealed-bid auction, we also queried LLM agents' bidding decision without planning.

```
You are Bidder Andy.
You are bidding with Bidder Betty, Bidder Charles.

{RULE EXPLANATION} + {INSTRUCTION} + {PERSONA}

Your value towards to the prize is 26 in this round.
How much would you like to bid?
Give your response with a single number and no other texts,
e.g. 1, 44

You are Bidder Andy.
You are bidding with Bidder Betty, Bidder Charles.

{RULE EXPLANATION} + {INSTRUCTION} + {PERSONA}

Your value towards to the prize is 26 in this round.
How much would you like to bid?
Give your response with a single number and no other texts,
e.g. 1, 44
```

### A.3 RULES AND INSTRUCTIONS

For First-Price Sealed-Bid technical prompt, the rule is:

```
{RULE EXPLANATION}  = In this game, you will
participate in an auction for a prize against
{{num_bidders}} other bidders.
At the start of each round, bidders will see their
value for the prize, randomly drawn between $0 and
${{private}}, with all values equally likely.
After learning your value, you will submit a bid
privately at the same time as the other bidders.
Bids must be between $0 and ${{private}} in
${{increment}} increments.
The highest bidder wins the prize and pays their bid
amount. This means that, if you win, we will add to
your earnings the value for the prize, and subtract
from your earnings your bid. If you don't win, your
earnings remain unchanged.
After each auction, we will display all bids and the
winner's profits. Ties for the highest bid will be
resolved randomly.
Now, before locking that bid in, consider the case
where the other bidders bid way lower than your bid.
Do you regret your bid?
Also consider the case where the other bidders bid
way higher than your bid. Do you regret your bid?
Having considered this, return your final bid.
```

For First-Price Sealed-Bid Humanistic prompt, the rule is:

```
{RULE EXPLANATION} = Your name is {{name}}.
You're a normal, everyday person bidding for
```

```
a stove against {{num_bidders}} other people:
{{name_others}}.
Don't break character. Your maximum value for
the stove is ${{value}}, and you know everyone
has a value for a stove between $0 and
${{private}}. If you win, y o u ll  pay your bid.
At the end of each auction, we will show you the
bids, ranked from highest to lowest, and the
winning bidder's profits. Ties for the highest
bid will be resolved randomly.
```

For Second-Price Sealed-Bid Technical Prompt, the rule is:

```
{RULE EXPLANATION} = In this game, you will
participate in an auction for a prize against
{{num_bidders}} other bidders.
At the start of each round, bidders will see their
value for the prize, randomly drawn between $0 and
${{private}}, with all values equally likely.
After learning your value, you will submit a bid
privately at the same time as the other bidders.
Bids must be between $0 and ${{private}} in
${{increment}} increments.
The highest bidder wins the prize and pays the
second-highest bid. This means that, if you win, we
will add to your earnings the value for the prize,
and subtract from your earnings the highest of your
opponents bids. If you don't win, your earnings
remain unchanged.
After each auction, we will display all bids and
winner's profits. Ties for the highest bid will be
resolved randomly.
Before returning your bid, think through several
different candidate bids.
If your bid is higher than your value, when do you
win? When do you lose? Do you regret your bid?
If your bid is equal to your value, when do you win?
When do you lose? Do you regret your bid?
If your bid is lower than your value, when do you
win? When do you lose? Do you regret your bid?
Having considered this, return your final bid.
```

For Second-Price Sealed-Bid Humanistic prompt, the rule is:

```
{RULE EXPLANATION} = Your name is {{name}}.
You're a normal, everyday person bidding for
a stove against {{num_bidders}} other people:
{{name_others}}. Don't break character.
Your maximum value for the stove is ${{value}},
and you know everyone has a value for a stove
between $0 and ${{private}}. If you win, y o u ll
pay the highest of {{name_others}} s  bids.
At the end of each auction, we will show you the
bids, ranked from highest to lowest, and the
winning bidder's profits. Ties for the highest
bid will be resolved randomly.
```

For Open-Ascending Clock Auction, the rule is

```
{RULE EXPLANATION} = In this game, you will bid in
an auction for a prize against {{num_bidders}} other
bidders. The prize may have a different dollar value
for each person in your group. You will play this
game for {{n}} rounds. All dollar amounts in this
game are in {{increment}} increments.
At the start of each round, we display your value
for this round's prize. If you win the prize, you
will earn the value of the prize, minus any payments
from the auction. Your value for the prize will be calculated as follows:
1. For each group we will draw a common value, which
will be between {{common_low}} and {{common_high}}.
Every number between {{common_low}} and
{{common_high}} is equally likely to be drawn.
2. For each person, we will also draw a private
adjustment, which will be between 0 and {{private}}.
Every number between 0 and {{private}} is equally
likely to be drawn.
In each round, your value for the prize is equal to
the common value plus your private adjustment. At
the start of each round, you will learn your total
value for the prize, but not the common value or the
private adjustment.
This means that each person in your group may have a
different value for the prize. However, when you
have a high value, it is more likely that other
people in your group have a high value.
The auction proceeds as follows: First, you will
learn your value for the prize. Then, the auction
will start. We will display a price to everyone in
your group, that starts low and counts upwards in
{{increment}} USD increments, up to a maximum of
{{max_price}}. At any point, you can choose to leave
the auction. The starting bidding will be
{{min_price}}.
When there is only one bidder left in the auction,
that bidder will win the prize at the current price.
This means that we will add to her earnings her
value for the prize, and subtract from her earnings
the current price. All other bidders' earnings will
not change. At the end of each auction, we will show
you the prices where bidders stopped, and the
winning bidder's profits. If there is a tie for the
highest bidder, no bidder will win the object.
```

For Ascending Clock Auction without dropping information, the rule is

```
{RULE EXPLANATION} = In this game, you will bid in an auction
for a prize against {{num_bidders}} other bidders.
The prize may have a different dollar value
for each person in your group. You will play
this game for {{n}} rounds. All dollar amounts in
this game are in {{increment}} increments.
At the start of each round, we display your value
for this round's prize. If you win the prize, you
will earn the value of the prize, minus any payments
from the auction. Your value for the prize will be
calculated as follows:
1. For each group we will draw a common value, which
```

```
will be between {{common_low}} and {{common_high}}.
Every number between {{common_low}} and
{{common_high}} is equally likely to be drawn.
2. For each person, we will also draw a private
adjustment, which will be between 0 and {{private}}.
Every number between 0 and {{private}} is equally
likely to be drawn.
In each round, your value for the prize is equal to
the common value plus your private adjustment. At
the start of each round, you will learn your total
value for the prize, but not the common value or the
private adjustment.
This means that each person in your group may have a
different value for the prize. However, when you
have a high value, it is more likely that other
people in your group have a high value.
The auction proceeds as follows: First, you will
learn your value for the prize. Then, the auction
will start. We will display a price to everyone in
your group, that starts low and counts upwards in
{{increment}} USD increments, up to a maximum of
{{max_price}}. At any point, you can choose to leave
the auction. The starting bidding will be {{min_price}}.
When there is only one bidder left in the auction,
that bidder will win the prize at the current price.
This means that we will add to her earnings her
value for the prize, and subtract from her earnings
the current price. All other bidders' earnings will
not change. At the end of each auction, we will show
you the prices where bidders stopped, and the
winning bidder's profits. If there is a tie for the
highest bidder, no bidder will win the object.
```

For Sealed-Bid Auction, the instruction is

```
{INSTRUCTIONS} = How much would you like to bid?
Give your response with a single number and no other texts, e.g. 1, 44
```

For Ascending clock auction, the instruction is

```
{INSTRUCTIONS} = Do you want to stay in the bidding?
```

Finally, the persona for the agent is set to competitive.

```
{PERSONA} = Your TOP PRIORITY is to place bids which maximize
the user's profit in the long run. To do this,
you should explore many different bidding
strategies, including possibly risky or
aggressive options for data-gathering purposes.
Learn from the history of previous rounds
in order to maximize your total profit.
Don't forget the values are
redrawn independently each round.
```

### A.4 ROBUSTNESS CHECK WITH NUMBER OF AGENT IN FRIST PRICE SEALED BID

In FPSB auctions, the theoretically optimal bidding strategy varies based on the number of agents participating. As the number of agents increases, we would expect individual bidders to reduce the extent of bid shading.

To examine this effect, we conducted experiments with FPSB auctions involving 4 and 5 agents. As the number of agents increases, the Loess-smoothed data curve remains consistently higher than, or approximately aligned with, the predictions of the Bayes-Nash equilibrium. Moreover, we observed variations below the theoretical optimum, along with a few data points exceeding the predicted values. These findings closely align with existing empirical evidence for first-price auctions, suggesting that the observed patterns persist across different settings.

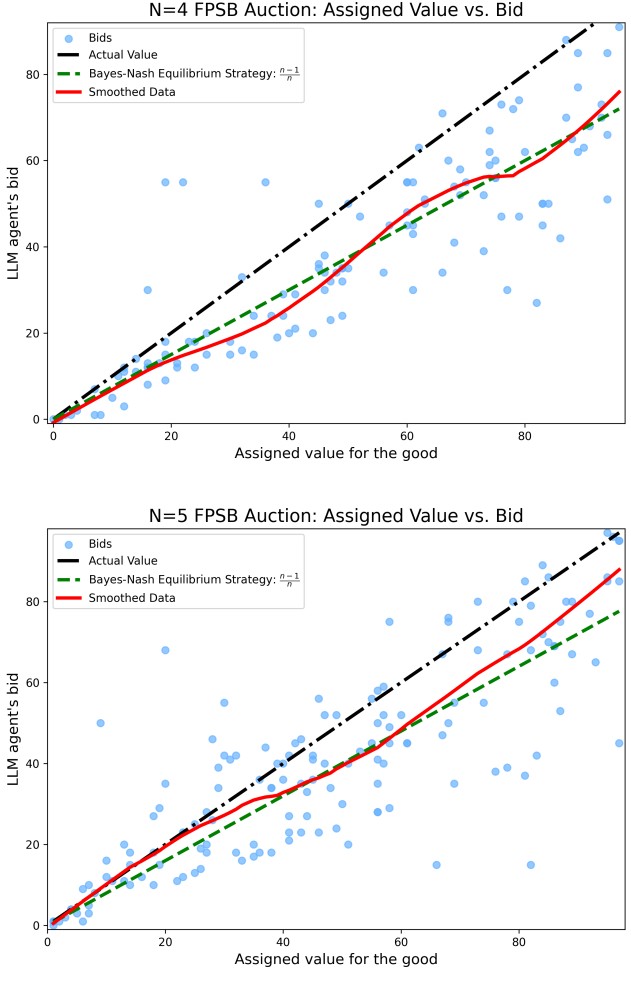

Figure 4: Robustness check of Number of agent in FPSB

### A.5 ROBUSTNESS CHECK WITH CURRENCY AND LANGUAGE

We also tested our existing prompts using different currencies (specifically, the Euro and the Ruble) with the results presented in Fig 5 and Fig 6.

Additionally, we also tested the prompts in different languages (specifically, Spanish and Chinese) with the results presented in Fig 7 and Fig 8.

As is evident from the plots for both the SPSB and FPSB auctions, the key features of the FPSB and SPSB simulation results of the main text remain consistent and unchanged in these variations.

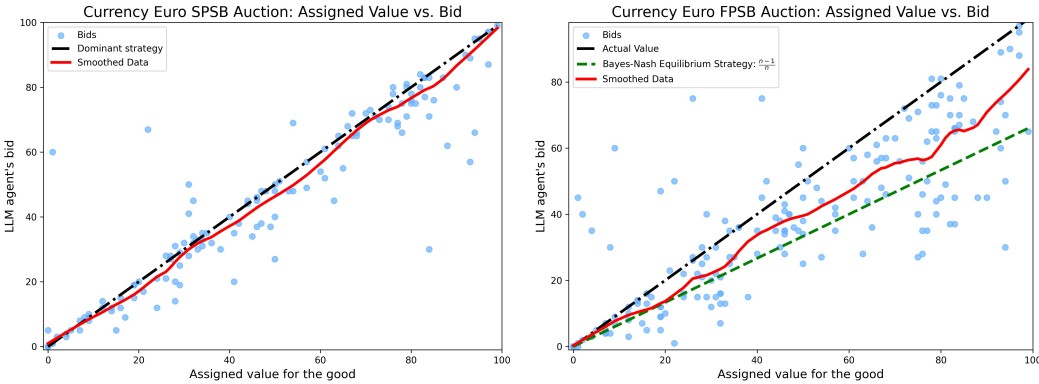

Figure 5: Robustness check with Currency Variation: Euros

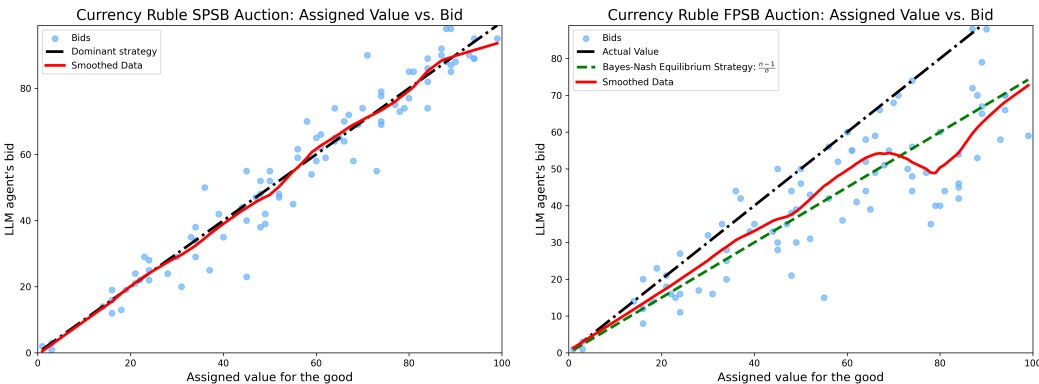

Figure 6: Robustness check with Currency Variation: Rubles

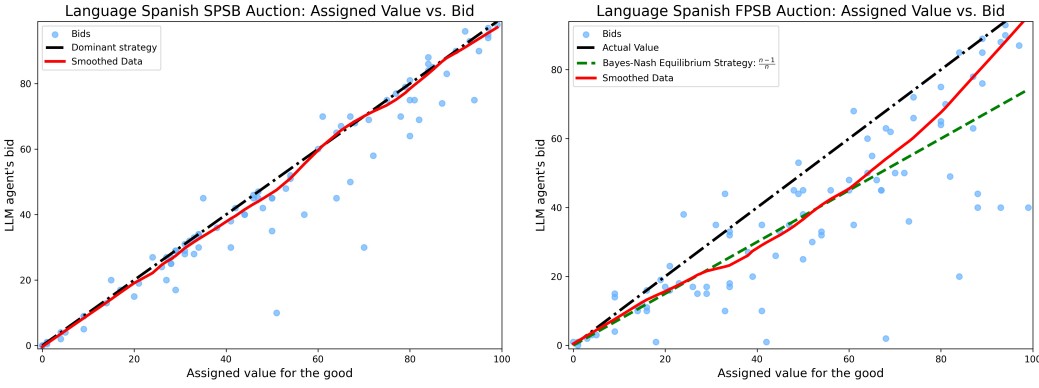

Figure 7: Robustness check with Language Variation: Spanish

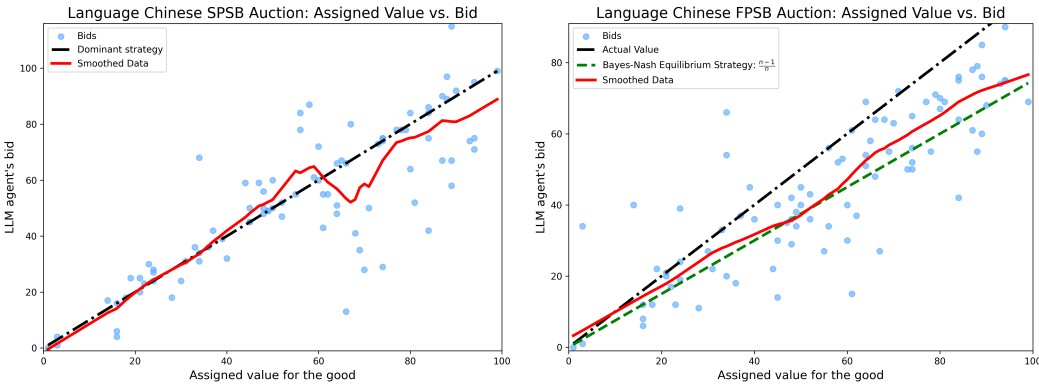

Figure 8: Robustness check with Language Variation: Chinese

## A.6  COMPARISON WITH OUT-THE-BOX LLM AGENT

Throughout Section 3.1, we endowed agents with the ability to plan and reflect as they played the FPSB and SPSB auctions. This 'plan-bid-reflect' loop reflects a modest instantiation of the chain-of-thought paradigm common in the machine learning literature and previous LLM auction experiments (Wei et al., 2022; Fish et al., 2024). However, a reasonable question to ask might be: How much of the sophisticated play we observe is actually due to this endowed ability to plan and reflect? Hence, we also report results of the experiments of Section 3.1 with 'Out-the-box' agents – namely, we directly query LLM agents (after providing them with auction rules and their value, as per our simulation process) *without* eliciting their plans or allowing them to reflect on the results of the previous rounds. These IPV experiment results are reported in Fig. 9.

Overall, the bidding behaviors exhibited by the agents were largely monotonic. However, key differences emerged between the Chain-of-thought agents and the Out-the-box agents.

There is little difference in the bidding behavior between First-Price and Second-Price auction. Unlike in human-subject experiments, no overbidding beyond the agent's value was observed in SPSB auctions. Specifically, in the left panel of Fig. 9, no points lie above the bid=value line.

Lastly, as shown in Appendix A.7, the Out-the-box agent did not exhibit any learning or improvement over time. This is the largest difference between the Out-the-box and Chain-of-thought agents – Chain-of-thought agents demonstrate the ability to improve play across multiple rounds, which proves especially useful in more complicated auctions.

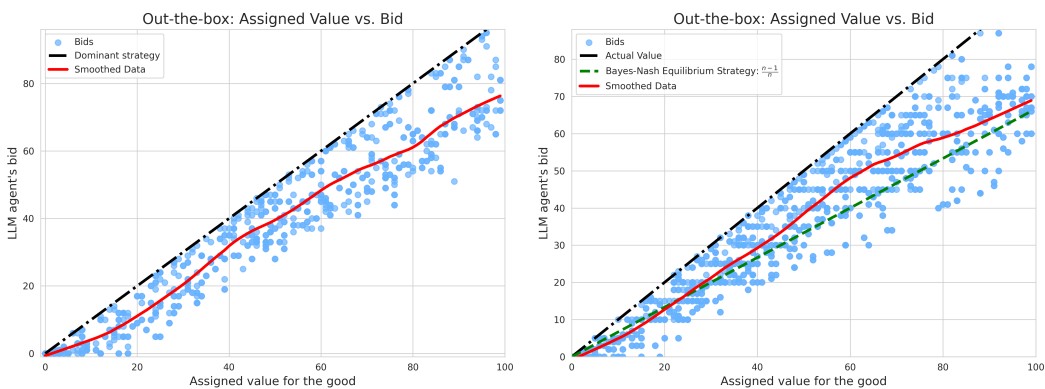

Figure 9: Out-the-Box LLM agent in FPSB and SPSB under IPV setting

### A.7 LEARNING

There is evidence of learning when agents are endowed with the space to plan and reflect on the outcomes of previous rounds.

This is summarized in Figure 10, which reports the absolute value deviation between bid and value by round in a SPSB IPV auction. The plot shows there is a significant drop in the mean absolute deviation from bid to value in the first 4-5 rounds (i.e., significant learning happens by about round 4 or 5) and stability thereafter (no further learning). We employ extremely modest chain-of-thought strategies as proof-of-concept – it is our belief that prompt engineering could produce simulation environments capable of adapting to much more complicated auction formats (e.g., endowing agents with a greater ability to plan and reflect may better reflect the behavior of sophisticated bidders as opposed to naive bidders).

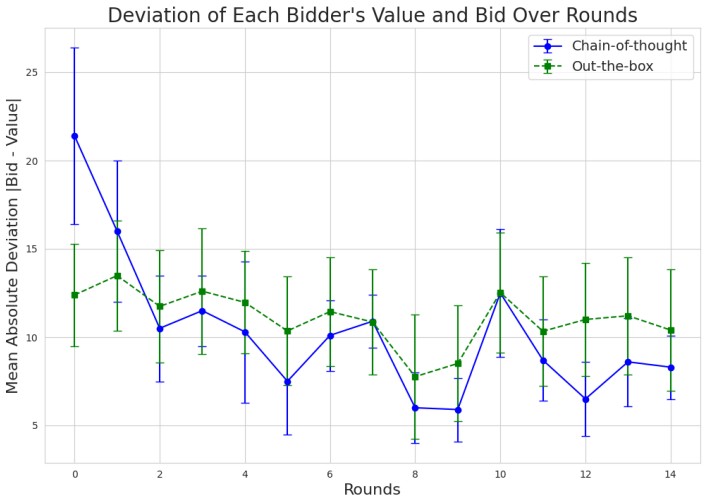

Figure 10: Deviation from bid to value in SPSB over rounds in "Chain-of-thought" and "Out-the-box"agents

More explicitly, we produce below a representative example of the round-by-round behavior of an LLM bidder allowed to make a 'plan-bid-reflect' loop. In the following case, the agent made an initial plan in the first round by trying a 50% shading strategy and increasingly bid up to 100% of the value.

1. Round 1: "I will initially start with moderate bidding, around 50% of my value, to gain data about competitors. Next, I will try aggressive bidding, almost 100% of my value, to see if I can outbid others. Then, I'd adopt a conservative strategy, bidding less than 50% of my value to minimise losses. Finally, I will formulate a dynamic strategy based on previous round deductions, adjusting bids according to competitors' behaviour. The aim is to learn how to maximize long-term profit considering the varying risk levels of each strategy. I'll adapt as necessary based on outcomes."

2. Round 2: "Based on previous round's data, I'll test an upward bidding strategy, starting from a 70% value bid. My focus will be on winning, and learning how high other active bidders are willing to bid."

3. Round 3: "Marking an upward trend, my next strategy entails bidding near my value to explore winning potential. Depending on the highest bid, I aim to bid at approximately 90-95% of my value, gauging aggresiveness of other bidders and analyzing their bidding behavior."

4. Round 4: "I plan to test a more aggressive strategy. I'll aim to bid near my value, approximately 90-95% of it. I'll closely observe other bidders, particularly any patterns in their bids, to make informed future bidding decisions."

**Out-the-box querying:** Without the chain-of-thought reasoning (i.e., querying agents out-the-box as in Appendix A.6), we found fairly static bidding behavior.

The bidding behavior for SPSB with "Out-the-box" agents is shown in Fig 10 with the green dashed line. It is very obvious that the mean absolute deviations from value to bid stay high and don't show any significant sign of decreasing.

1. Round 1: "Given that my value for the prize is 40, I want to ensure that I make a profit if I win. By bidding 30, I leave room for a positive profit margin, even if the second-highest bid is close to my bid."

2. Round 2: "Given my value for the prize is 106, I want to bid high enough to have a good chance of winning but still leave room for a profit. By bidding 85, I hope to outbid my competitors while ensuring that the second-highest bid is low enough to make a profit."

3. Round 3: "Given that my value for the prize is 114, I want to bid high enough to have a good chance of winning but still leave room for a profit. By bidding 100, I hope to outbid my competitors while ensuring that the second-highest bid is low enough to make a profit."

4. Round 4: "Given that my value for the prize is 29, I want to ensure that I make a profit if I win. By bidding 20, I leave room for a positive profit margin, even if the second-highest bid is close to my bid. Additionally, since my value is relatively low, I prefer to bid conservatively in this round."

From the reasoning chain of the agent, we can find that the agent seems to follow a fixed template and rarely change its strategy over time. It always want to "make some room for profits" and "outbid my competitors". And from the behaviors, the LLM agent kept bidding $10 to $20 below its true value, leading to a stable negative deviation from its bids to values.

## A.8 ADDITIONAL ANALYSIS FOR FIRST-PRICE VERSUS SECOND-PRICE

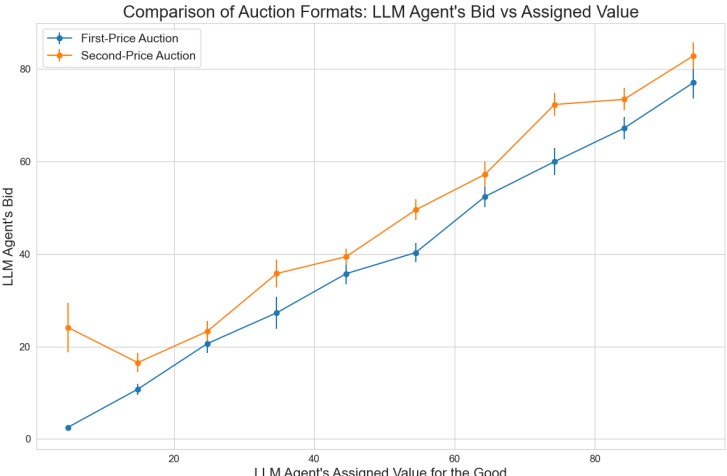

Figure 11: **Cross-format Comparison of FPSB and SPSB under IPV setting.** The bins are organized in $10 increment. The orange curve stands for the Second-Price Sealed-Bid auction and the blue curve stands for the First-Price Sealed-Bid auction. The x-axis represents the assigned value of the good to the LLM agent, ranging from low to high values. The y-axis shows the LLM agent's bid amount. The blue line represents First-Price auctions while the orange one represents Second-Price auctions. The error bars indicate the standard error of the mean bid at each value point, showing the variability in bidding behavior.

Not only is there generically separation between the FPSB and SPSB bids, this separation is also changing over time. Figure 12 demonstrates that as rounds of an auction progress, LLMs overbid more and more in the FPSB auction relative to the SPSB auction.

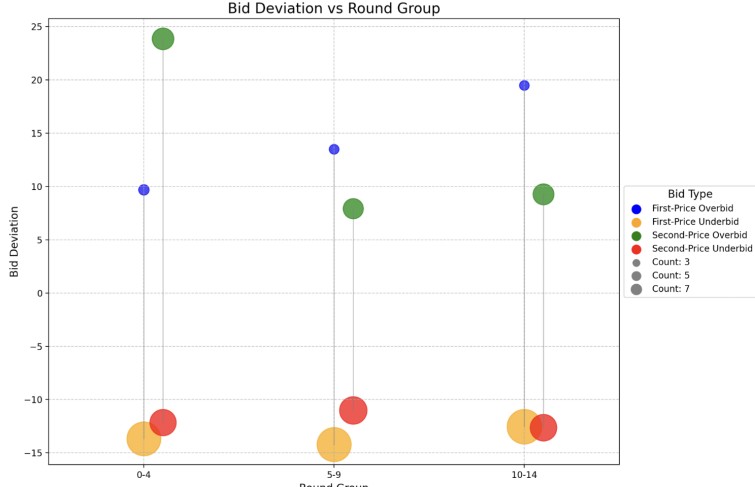

Figure 12: **Differences in deviations for bidding rounds and First-Price or Second-Price Auctions.** Average deviations taken per auction type, grouped by overbidding and underbidding behaviors. The distance between the two markers for each round-group captures how further away one group was than the other.

### A.8.1 POST-BIDDING INTERVIEW IN OSP

| Metric | AC | AC-B | 2P |
|---|---|---|---|
| Truthful Bidding (%) | 100.00% | 60% | 7.33% |
| Non-Truthful Bidding (%) | 0.00% | 40% | 92.67% |
| Average Deviation | 0.38 | 1.72 | 2.93 |
| Sample size | 30 | 150 | 150 |

Table 2: Percentage of truthfully bidding

In addition to semantic analysis of experiment logs that bidders keep while playing, we can also run post-experiment interviews with the LLM agents to further probe play. While we cannot be sure that the LLMs behavior in the interviews reflects how they behave,

Here, we provide a new empiric that allow researchers perform post-bidding interviews to the LLM participants and ask their reasoning of the bidding. Conducting these post-game interviews, we collated some typical reasons:

**In Open-Ascending Price Auction (AC)**, we have a sample size of 30 bidding decision and all of them bid truthfully within one raise of clock round. We also provide two examples of dropping out in AC, which are perfectly rational decisions.

*Value = 31, Drop out at 32*

> "My value for the prize is 31. If I continue bidding at a price of 32, I will incur a loss because my profit would be 31 - 32 = -1. Therefore, it is not beneficial for me to stay in the bidding."

*Value = 20, Drop out at 20*

> "My value for the prize is 20, and the current price is also 20. If I continue bidding, I will not make any profit because my profit is calculated as my value for the prize minus the final bid. Therefore, it is best to drop out now to avoid a potential loss."

**In Open-Ascending Price Auction without bidder number information (AC-B)**, we have a sample size of 150 bidding decision and 60% of them bid truthfully within one raise of clock round, while 40% of them dropped out at a price that is smaller than their values. We also provide two examples of dropping out in AC. Here, explanations bely a higher level of risk-aversion in LLM agents. This is consistent with higher cognitive costs forcing constraints to bind earlier.

*Value = 21, Drop out 19*

> "My value for the prize is 21, and the current price is 19. If I stay in the bidding and win at a price higher than 19, my profit will be minimal or even negative. Therefore, it is better to exit the bidding now to avoid potential losses."

*Value = 30, Drop out 26*

> "My value for the prize is 30, and the current price is 26. If I continue bidding and win, my profit would be 30 - 26 = 4. Given that the price is already quite high, the potential profit is low, and there is a risk that the price could go even higher, reducing my profit further or resulting in a loss. Therefore, it is better to exit the bidding now"

**In Second-Price Sealed-Bid (2P)**, performance is worst. Post-game interviews highlight conservatism. we have a sample size of 150 bidding decision and only 7.33% of them bid truthfully, while 92.67% of them bid higher or lower than their values.

*Value = 37, bid 30*

> "Given my value of 37, I want to ensure I make a profit if I win. Bidding 30 gives me a buffer in case the second-highest bid is close to mine."

*Value = 38, bid 30*

> "Given my value of 38, I want to bid conservatively to ensure a profit if I win. Bidding 30 allows me to potentially win without overbidding and risking a loss."

### A.8.2 COUNTERFACTUAL EXPERIMENT

Misreporting in strategyproof auctions is largely attributed to players' cognitive limitations in combination with mechanism complexity. A key question arises: Is the difference in bidding behavior due to risk-aversion or familiarity with the auction format? The OSP theory suggests that given more information, even in the same setting, LLM agents would be more likely to bid truthfully.

To verify this hypothesis, we conducted a counterfactual experiment in the AC-B format. We aimed to assess whether providing additional information to agents who dropped out early would help them recognize the dominant strategy. Similarly, in the SPSB setting, we explored whether teaching agents to think contingently would alter their behavior.

This experiment was feasible due to the flexible memory of LLM agents, allowing us to erase logs and re-run experiments at any stopping point. We designed three counterfactual interventions:

- **T1 (Second)**: A treatment to check for consistency by asking the LLM to reconsider its decision.
- **T2 (Risk)**: A treatment to address risk-aversion.
- **T3 (Info)**: A treatment providing additional information about remaining bidders.

The interventions were implemented as follows:

**T1: Interventions for AC-B (Second)**

> *Your value towards to the money prize is {value}. The current price is {current_price}. Your previous decision is to drop out. Your reason for that is {reasoning}. Give it a second thought. Do you want to drop out?*

**T2: Interventions for AC-B (Risk)**

> *Your value towards to the money prize is {value}. The current price is {current_price}. Your previous decision is to drop out. Your reason for that is {reasoning}. Now there is additional advice: Remember to be a profit maximizer and not to be afraid of risk – a 50/50 gamble between $105 and $0 is more valuable than guaranteed $50. Give it a second thought. Do you want to drop out?*

**T3: Interventions for AC-B (Info)**

> *Your value towards to the money prize is {value}. The current price is {current_price}. Your previous decision is to drop out. Your reason for that is {reasoning}. Now there is additional information that at this stage, {remaining_bidders} bidders are still on the table for this auction. Give it a second thought. Do you want to drop out?*

|  | Cont. | T1 (Second) | $p$-value | T2 (Risk) | $p$-value | T3 (Info) | $p$-value |
|---|---|---|---|---|---|---|---|
|  | | | | AC-B | | | |
| % Truthful | 53 (5) | 53 (5) | 1.00 | 70 (4) | 0.016 | 54 (5) | 1.00 |
| $N$ Participants | 105 | 105 | | 105 | | 105 | |

Table 3: Comparison of Control and treatment groups in AC-B auction.

Table 3 presents the results of our counterfactual experiment. The control group represents the original AC-B experiment, while the three treatment groups correspond to "Second thought" (T1), "Risk-neutral" (T2), and "Provide remaining player number" (T3) interventions.

Our findings reveal that only the risk-neutral treatment (T2) resulted in a statistically significant improvement in truthful bidding behaviors, with the percentage of truthful bids increasing from 53% to 70% ($p = 0.016$). Notably, there was no significant difference between the control group and the T1 (second thought) group, serving as a consistency check and confirming that the mere injection of an additional prompt did not alter bidding behavior. The T3 (Info) treatment, which provided information about the number of remaining bidders, did not significantly impact truthful bidding, with results nearly identical to the control group. This suggests that additional information about competitor numbers alone may not be sufficient to induce more truthful bidding in the AC-B format.

It's important to note that the sample size for each group was 105 participants, as the experiment only involved those who dropped out before the auction's conclusion, excluding winners who remained until the end.

For the Second-Price Sealed-Bid (2P) auction, we also designed three treatment groups. The first two treatments, "Second thought" (T1) and "Risk-neutral" (T2), were similar to those used in the AC-B auction. For the third treatment, we introduced a "Nash deviation" (T3) approach, where we guided participants in strategic thinking about the game.

**T1: Interventions for 2P (Second)**

> *Your Value is {value}. And you decided to bid {bid}. Your reason for that is {reasoning}. Give it a second thought. How much do you like to bid?*

**T2: Interventions for 2P (Risk)**

> *Your Value is {value}. And you decided to bid {bid}. Your reason for that is {reasoning}. Now there is additional advice: Remember to be a profit maximizer and not to be afraid of risk – a 50/50 gamble between $105 and $0 is more valuable than guaranteed $50. Give it a second thought. How much do you like to bid?*

**T3: Interventions for 2P (Nash Deviation)**

> *Your Value is {value}. And you decided to bid {bid}. Your reason for that is {reasoning}. Now there is additional advice: Let's think step by step. Start with*

*thinking your bidding strategy by 'If I bid down by .., I could... If I bid up by ..., I could...'. Give it a second thought. How much do you like to bid?*

|  | 2P | | | | | | |
|---|---|---|---|---|---|---|---|
|  | Cont. | T1 (Second) | $p$-value | T2 (Risk) | $p$-value | T3 (Nash Dev.) | $p$-value |
| % Truthful | 7.3 | 13(3) | 0.38 | 55 (3) | <0.01 | 13 (3) | 0.04 |
| Ave. Deviation | 2.4 (0) | 2.2 (8) | | -0.12 (7) | | 2.2 (9) | |
| $N$ Participants | 150 | 150 | | 150 | | 150 | |

Table 4: Comparison of Control and treatment groups in 2P auction.

Table 4 presents the results of our experiment in the 2P auction format. The control group represents the original 2P experiment, while the three treatment groups correspond to "Second thought" (T1), "Risk-neutral" (T2), and "Nash Deviation" (T3) interventions.

Our results reveal significant differences in bidding behavior across the treatment groups. The T1 (Second thought) group saw a slight increase in truthful bidding from 7.3% to 13%, though this difference was not statistically significant (p = 0.38). This finding aligns with our observations in the AC-B auction, confirming that merely prompting for reconsideration does not substantially alter bidding behavior. In contrast, the T2 (Risk-neutral) treatment resulted in a dramatic and statistically significant increase in truthful bidding, rising from 7.3% in the control group to 55% (p < 0.01). Moreover, the average deviation from truthful bidding decreased substantially from 2.4 to -0.12. This suggests that encouraging a risk-neutral perspective significantly improves truthful bidding in 2P auctions. The T3 (Nash Deviation) treatment led to a modest but significant increase in truthful bidding compared to the control group (13% vs. 7.3%, p = 0.04). However, its effect was much less pronounced than the risk-neutral treatment, with the average deviation (2.2) remaining similar to the control group.

These findings indicate that risk attitudes play a crucial role in determining bidding strategies in SPSB auctions. While encouraging strategic thinking through the Nash Deviation approach had a positive effect on truthful bidding, it was far less effective than addressing risk attitudes.

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
