# OpenReview forum: "Evidence from the Synthetic Laboratory: Language Models as Auction Participants"
_ICLR.cc/2025/Conference — ICLR 2025 Conference Withdrawn Submission_

### Official Review · Reviewer_qbpD · 2024-10-28

**Soundness:** 2
**Presentation:** 3
**Contribution:** 2
**Rating:** 5
**Confidence:** 4

**Summary:**

This paper studies the auction behavior of LLMs. They benchmark LLM agents against two kinds of existing experimental results. The first result is that realized human biddings differ from theoretical results in first-price and second-price auctions. The second result is that clock auctions induce more rational play in humans than second-price auctions. The authors systematically designed prompts and their results show that the behavior of LLMs conforms with the aforementioned experimental results.

**Strengths:**

1. The study of LLMs in auctions are interesting and relatively new.

2. In the appendix, the authors include robustness checks, which make the results more convincing.

3. The main structure of the paper is clear. Each benchmark begins with the setting, proceeds to existing theoretical and empirical results, and ends with simulation outcomes, which make it easy to be understood.

**Weaknesses:**

1. My main concern is that the technique contribution of this paper is limited. The method and design of experiments are simple and not novel.

2. The results claimed in abstract are not clearly presented.
  * The authors claim that
>Our results also suggest that LLMs are bad at playing auctions ‘out of the box’ but can improve their play when given the opportunity to learn.
>
   On one hand, it not clear which part of their results shows that LLMs are bad at playing auctions ‘out of the box’. On the other hand, in sec 3.2.4, they mention that
>Interestingly, we see little evidence of learning over time.
>
This seems to contradict with the conclusion in the abstract and I think clarification is needed.

* The authors claim that
>..., validating a novel synthetic data-generating process to help discipline the study and design of auctions.
>
And the authors actually ask two research questions:
>The present work examines this question for auctions: how well do LLMs substitute for humans in auctions, especially when behavioral traits like risk-aversion or bounded rationality matter? And, more generally: how can we use LLMs to improve the design of economic mechanisms?
>
The first question is answered well. But it is not clear that which part of their results discusses using this process to improve the design of economic mechanisms.

Overall, I suggest the authors summarize their research questions and contributions in a more explicit way.

**Questions:**

1. Why the authors use LOESS-smoothed data for regression? Is it standard? I think a few explanations are needed here.

2. In sec 3.2.4, the auctions repeat 15 rounds and the LLM agents seem not improve their understanding of the mechanisms by bidding closer to their true value over time. Did the author try to repeat the auction more than 15 rounds?

---

> ### Author Response · Authors · 2024-12-04
> **Learning**
>
> _"On one hand, it not clear which part of their results shows that LLMs are bad at playing auctions ‘out of the box’. On the other hand... their writing contradicts with the conclusion in the abstract and I think clarification is needed."_
>
> Thank you for the comment. We agree the previous draft could have been clearer on learning, and we hope you find that the updated draft more satisfactory in this regard. We have fixed the language in this section to more clearly emphasize that we observe learning behavior for LLM agents endowed with the ability to plan and reflect.
>
> Additionally, we've also ran additional experiments with `Out-the-box' agents in the updated draft to demonstrate more precisely the difference in learning between `Out-the-box' agents and `Chain-of-thought' agents. These additional experiments are reported in Appendix A.7.

---

### Official Review · Reviewer_MWn6 · 2024-11-02

**Soundness:** 3
**Presentation:** 3
**Contribution:** 3
**Rating:** 6
**Confidence:** 3

**Summary:**

This paper investigates the auction behavior of simulated AI agents (LLMs) and validates a synthetic data-generating process for auctions.
Precisely, this paper benchmarked LLM agents against established experimental results in auctions, such as revenue equivalence between first-price and second-price auctions and behavior in obviously strategy-proof auctions. It is found that when LLMs diverge from theory, their behavior aligns with some human behavioral traits observed in experimental economics literature (e.g., risk aversion). Also it is shown that LLMs can improve their play with learning opportunities and this learning is robust across settings.

**Strengths:**

1. Provides insights into how LLMs perform in auction scenarios, which is relevant as LLMs are increasingly being considered for various economic applications.
2. Conducts a detailed analysis of LLMs' behavior in different auction formats and settings, including semantic analysis and counterfactual experiments to understand their decision-making processes.

**Weaknesses:**

1. "Each setting is repeated 15 times with values drawn randomly each time." It is hard to say whether 15 samples are enough to represent a random distribution.
2. It is a pity there is no theoretical result can be derived or developed based on the experiments.

**Questions:**

NA

---

> ### Author Response · Authors · 2024-12-04
> **Robustness**
>
> _"Each setting is repeated 15 times with values drawn randomly each time." It is hard to say whether 15 samples are enough to represent a random distribution._
>
> Thank you for the comment. In the updated draft, we incorporated the results of >100 new runs and found our results unchanged. We also hope to make simulation runs with more than 15 rounds in the near future.

---

### Official Review · Reviewer_EvBN · 2024-11-05

**Soundness:** 3
**Presentation:** 3
**Contribution:** 3
**Rating:** 6
**Confidence:** 4

**Summary:**

This paper studies how well LLMs can simulate human behavior in auctions. The key motivation behind this work is that generating synthetic data using LLMs could potentially offer a cheaper alternative to traditional human-subject experiments.

The findings show that LLMs sometimes deviate from the predictions of economic theory, but these deviations often align with the behavioral traits observed in real-world experiments with human participants. In particular, the LLMs tend to underbid in second price auctions and overbid in first price auctions, yielding a counter argument to revenue equivalence. This can be explained by risk aversion in LLM bidding behavior. Additionally, the paper reveals that LLMs perform better in simpler auction formats (clock auctions), which supports the argument behind the obvious strategy-proofness.

The authors suggest that LLMs could be a valuable tool for studying a wide range of economic mechanisms, including those that are currently difficult or impossible to test experimentally.  They highlight the potential of LLMs to generate data for large-scale experiments that would be too costly or raise ethical concerns if conducted with human participants.

**Strengths:**

This paper moves along an exciting application of LLMs in experimental economics, and could be a valuable and unique plus to ICLR. The economic properties being examined in this paper are fundamental and the observations from LLMs are interesting.

**Weaknesses:**

One major challenge to this line of research (LLMs as an alternative to human-subjects in experiments) is that the behavior of the LLMs could be highly sensitive to (1) training data and (2) prompt, which may put the reliability of observations from experiments with less known results at risk. More importantly, the complete prompts used in this work are not shared and there seems no robustness test of the LLM behavior against the prompts.

Taking the first prompt template in Appendix A.1 as an example, where “RULE EXPLANATION”, “INSTRUCTIONS”, “PERSONA” are not given. I used this template by only filling the rule explanation part and observed very different results (i.e., the bidding plan from the LLM).
* If I describe the auction rule of second price auction in a less formal way (in the tongue of ordinary people), the LLM replies things like slightly overbidding and randomized bidding etc.
* If I describe the auction rule of second price auction in a formal way (in the tongue of an auction theorist, e.g., using “sealed bid second price auction”), the LLM immediately says “my dominant strategy is to bid my true valuation of the item”.

From the simple test above, we can see that response from the LLM could depend on (1) whether they learned how to play this game from training data (they may know the literature of the experiment already) and (2) whether the prompt properly recalls the corresponding memory.

Therefore, a crucial component of studies like this would be a robustness test.

**Questions:**

What are the prompts used as “RULE EXPLANATION”, “INSTRUCTIONS”, “PERSONA” in this work?

---

> ### Author Response · Authors · 2024-12-04
> **Robustness**
>
> Thank you for the comment. In the updated draft, we've included multiple new robustness checks in Appendix A.4 and A.5.

---

### Official Review · Reviewer_E3Av · 2024-11-08

**Soundness:** 3
**Presentation:** 3
**Contribution:** 3
**Rating:** 8
**Confidence:** 3

**Summary:**

This manuscript asks, "how well do LLMs substitute for humans in auctions"? - a question motivated by the cost of experimenting with human subjects: the c. 1,000 auctions run for the manuscript cost less than US\\$100 on GPT-4 and GPT-4o - in contrast to the US\\$15,000 price of the human experiments (with 404 subjects) in Li (2017).

The paper first describes the LLM simulation environment (§2).  Then §3 forms the bulk of the paper, introducing the auction designs (first and second price sealed bid - FPSB and SPSB - as well as the strategically equivalent ascending auction, AC, and a 'blind' variant, AC-B, which does not report when bidders drop out).  For each, theoretical and (human) experimental results are presented.  These are compared to the LLM results.

The findings are:
1. consistent with theory, LLMs bid higher in the SPSB than in the FPSB, although with a "smaller separation ... than would be predicted by theory" - largely due to overbidding relative to theory in the FPSB;
1. in AC - which is 'simpler' than the strategically equivalent auction SPSB - LLM play is closer to theoretically predicted play.  This is found in both independent private values (IPV) and affiliated private values (APV) settings.

**Strengths:**

**originality**

The manuscript is original in the sense of being the first I have seen to assess LLMs as human substitutes in auctions.

**quality**

The research is well reasoned, conducted and written up.

**clarity**

The manuscript is generally clear.

**significance**

To use the expression of Dell'Acqua et al. (2024), the capabilities of LLMs form a 'jagged frontier' relative to human performance, requiring task-by-task analyses.  This manuscript contributes to that endeavour.

**Weaknesses:**

1. above all, I wonder what problems are solved by the discovery that LLMs in stylized environments play roughly like humans do.  Thinking aloud, some guesses:
   1. testing failure modes in advance of high-profile auctions.  Even given the relatively low costs of LLM use, this seems an inefficient way of proceeding relative to e.g. randomly generating bid data or formally proving properties (q.v. Caminati, Kerber, Lange and Rowat, 2015).  To be convinced, I would want to see how well LLMs perform in more asymmetric auctions, e.g. calibrated to real high-profile auctions, or perhaps the Combinatorial Auction Test Suite (Leyton-Brown, 2024).
   1. assistance in gaining intuitions in theoretical analysis of novel auction formats (q.v. Dütting, Feng, Narasimhan, Parkes, and Ravindranath, 2024).  Here, the manuscript would be more convincing if it showed us insights into novel auction formats - instead of just the most common ones.
1. It is claimed (p.2) that LLMs may exhibit risk-aversion even when prompted not to.  The evidence for this claim is in Appendix A.4, which is not part of the submission.  Thus, I am unable to assess this claim.  I would, in particular, like to see it compared to other hypotheses.
1. It is claimed (p.3) that Li (2017) found "that human subjects tend to be more truthful in second price sealed bid auctions than in ascending clock auctions".  Again, I would want to see which other hypotheses were considered by Li: one can imagine sunk cost arguments affecting ascending clock bids, but not SPSB.  (A similar comment arises on p.8, when it is claimed that humans "are less truthful" under AC-B: alternatively, might they not just have more difficulty with the Bayesian calculation?)
1. It would be nice to see performance considered for varying $n$.  The manuscript indicates that $n$ is "often" three (p.3) or always three (p.4), which - I would guess - may account for around half of the examples in an LLM's training data.  Thus, it is possible that performance deviates from human considerably outside of the training data.
1. Other interesting robustness tests could include:
   1. compare results for auctions identical up to the 'currency' units (e.g. dollars, billions of cowrie shells, etc.).
   1.  compare results with and without the first step of the simulation procedure (p.3), which asks the LLM to generate a bidding plan explicitly: which result is closer to amateur/expert human bidding?
1. I would probably cite Vickrey's original paper for the results on his eponymous auction, rather than Krishna's 2009 textbook (p.5).
1. The panels in Figure 1 are reversed: that on the left, for which the bid = value in theory, is the SPSB, not the FPSB.
1. Given the importance of Li's OSP (2017), I would like a bit more space given to its explanation (e.g. maybe removing the table at the bottom of p.4, if needed).

**Questions:**

1. footnote 5 mentions setting a non-zero temperature to induce variation.  _Prima facie_, this might seem to correspond to something other than a BNE - like a trembling hand, or QRE?  (If so, than the comparison to BNE theoretical results is inappropriate.)
1. valuations in the FPSB and SPSB are drawn uniformly, but bids are constrained to be integers (p.4).  In general, equation (2), the FPSB BNE bid, does not resolve to an integer.  Is there any reason to believe that whatever rounding strategy is used does not introduce artefacts into the results?
1. there is evidence that, with training, humans learn to play close to Nash (pp.5, 8).  While the abstract claims that LLMs "can improve their play when given the opportunity to learn", the text (p.8) finds, "little evidence of learning over time".  Which of these is correct?  Either way, it may be useful to be more careful to compare like-for-like humans and LLMs (e.g. amateur/untrained v experienced/trained).

---

> ### Author Response · Authors · 2024-12-04
> **Framing / clarifications**
>
> Thank you for the close reading and the many insightful comments. Responses below.
>
> We welcome the comments on framing and exposition.  In the updated draft, we reworked the explanation of OSP mechanisms as you suggested and give a more thorough discussion interpreting the results in between the benchmarks of theoretical prediction and the existing human experimental literature. We also agreed the Vickrey citation was more natural for the SPSB results and appreciated the catch on the panels being flipped in Fig. 1. We also incorporated your recommendations on robustness checks and learning into the larger comments to all the reviewers above.

---

> ### Author Response · Authors · 2024-12-04
> **Testing other auction formats and comparison with theory.**
>
> *“1. testing failure modes in advance of high-profile auctions. Even given the relatively low costs of LLM use, this seems an inefficient way of proceeding relative to e.g. randomly generating bid data or formally proving properties (q.v. Caminati, Kerber, Lange and Rowat, 2015). To be convinced, I would want to see how well LLMs perform in more asymmetric auctions, e.g. calibrated to real high-profile auctions, or perhaps the Combinatorial Auction Test Suite (Leyton-Brown, 2024).*
>
> *2. assistance in gaining intuitions in theoretical analysis of novel auction formats (q.v. Dütting, Feng, Narasimhan, Parkes, and Ravindranath, 2024). Here, the manuscript would be more convincing if it showed us insights into novel auction formats - instead of just the most common ones.”*
>
> Thank you for the comments.
>
> We view our work as useful in accelerating empirical work (even if synthetic data isn't a perfect substitute for human empirics, it can help triage over which experiments to run) and as a complement to good theoretical work (quickly testing ideas). We wholeheartedly agree that the value of empirics is limited without theory for novel design, especially in auction theory / mechanism design work more broadly.
>
> Additionally, we agree that running simulations in asymmetric / novel auction formats are a natural next step. We've already run many simulations in combinatorial settings and are excited to see where that work leads. In this work, we report results for the most common auctions as we believe they are first order.

---

> ### Author Response · Authors · 2024-12-04
> **Risk-aversion**
>
> _"It is claimed (p.2) that LLMs may exhibit risk-aversion even when prompted not to. The evidence for this claim is in Appendix A.4, which is not part of the submission. Thus, I am unable to assess this claim. I would, in particular, like to see it compared to other hypotheses."_
>
> Thank you for the comment. In the updated draft, we include results running counterfactual experiments which prompt agents to be less risk-averse (also as compared to other hypotheses -- 'understanding', or how well the LLM seems to understand the mechanism, and 'reactiveness', or how well the LLM responds to non-equilibrium bidding from other agents). This work is in Appendix A.8.2.

---

> ### Author Response · Authors · 2024-12-04
> **Sunk cost fallacy**
>
> _“I would want to see which other hypotheses were considered by Li: one can imagine sunk cost arguments affecting ascending clock bids, but not SPSB. (A similar comment arises on p.8, when it is claimed that humans "are less truthful" under AC-B: alternatively, might they not just have more difficulty with the Bayesian calculation?)”_
>
> Thank you for the comment. Though we were unable to respond directly in this draft, we have two thoughts on how to respond to this in future work:
> 1) Experiments that prompt agents not to fall victim to the sunk cost fallacy (a la the method in Appendix A.8.2).
> 2) Experiments that `speed up’ or `slow down’ the experiment to LLM agents (tell them X minutes have elapsed between rounds, run rounds with larger or smaller time increments, etc). Such experiments might be particularly interesting to discipline by experiments with real people as well.

---

> ### Author Response · Authors · 2024-12-04
> **Solution concept -- BNE alternate?**
>
> _"footnote 5 mentions setting a non-zero temperature to induce variation. Prima facie, this might seem to correspond to something other than a BNE - like a trembling hand, or QRE? (If so, than the comparison to BNE theoretical results is inappropriate.)"_
>
> Thank you for the interesting comment, we enjoyed it. We agree that increasing temperature leads to more varied plans which in turn creates more variance in bidding relative to values. However, we view this as a feature rather than a bug - it allows us to study how LLMs reason about and learn optimal bidding strategies over multiple rounds, similar to how human subjects explore different strategies in experimental settings. The analogy is to an evolutionary process: we just want enough variation in plans to make learned, convergent behavior at the end all the more convincing.
>
> That being said – it is interesting to consider other solution concepts as theoretical benchmarks for our work. We intend to run some simulations in the next few weeks targeting the difference between solution concepts more directly and we welcome the comment.
>
> More substantively – we agree QRE may be an interesting way to think about our approach, as agents may choose suboptimal strategies from noise early due to an information dearth (some further evidence to this -- when instructed to reason through the SPSB dominant strategy equilibrium, LLMs play the SPSB much better). Trembling hand probably isn’t how we think of it, because the differences in plans are substantial enough that it's not due to noise appended to the ‘optimal plan’.

---

> ### Author Response · Authors · 2024-12-04
> **Rounding in LLM bids**
>
> _"valuations in the FPSB and SPSB are drawn uniformly, but bids are constrained to be integers (p.4). In general, equation (2), the FPSB BNE bid, does not resolve to an integer. Is there any reason to believe that whatever rounding strategy is used does not introduce artefacts into the results?"_
>
> Thank you for the comment.
>
> 1. We wanted to avoid using decimal numbers as there are some examples of GPT-4o being bad at making comparisons between them (e.g., [GPT4o thinking 1.11 > 1.9.](https://www.reddit.com/r/ChatGPT/comments/1e5fx67/ai_cannot_compare_decimals)). Even if it's a minor engineering fix, between that and the increased token cost we chose not to worry about it in this work.
> 2. There is also some use of rounding in prior empirical work. E.g., Kagel & Levin (1983) and Li (2017) both ask human participants to round their bid to the nearest $0.25.
>
> That being said, we appreciate the point. We'll run robustness tests on this in the next few weeks.

---

### Author Response · Authors · 2024-11-28
**Robustness checks**

We welcome the suggestion to contribute further robustness checks and thank the reviewers for comments that improved the paper. Inspired by reviewer comments, we have run the following three sets of additional experiments for robustness:

1. Comparing play under the FPSB and SPSB auctions (in IPV settings) along currencies other than the dollar (for now, the euro and ruble). We were pleased to see that changing the currencies did not suggest any serious change in our results (and in fact, a greater separation between FPSB and SPSB play than was reported in the main text).

More specifically:
- For the variation with euros instead of dollars, see Fig 5 in the updated draft.
- For the variation with rubles instead of dollars, see Fig. 6 in the updated draft.
- For context, the original plots from Section 3.1 are in Fig.1.

These are the only two currency variations we ran. Both are reported in Appendix A.5.

2. Additionally, we also compared play under the FPSB and SPSB auction in languages other than English (for now, Spanish and Chinese). Once again, we were pleased to see that changing the language did not suggest any change in our original results.

- For the variation with Spanish instead of English prompts, see Fig 7 in the updated draft.
- For the variation with Chinese instead of English prompts, see Fig 8 in the updated draft.

These are the only two language variations we ran. Both are reported in Appendix A.5.

3. Running our existing prompts along various numbers of agents (4 and 5 agents). We only ran these simulations for the first-price auction setting, as the BNE prediction in the FPSB depends on N. Extensions could also report results for other auction settings where the equilibrium prediction changes on N (e.g., third-price auctions, all-pay auctions, etc). We were pleased to see that LLM bids actually increased as N increased! I.e., evidence corroborating the BNE prediction of (N-1)/N increasing in N.

- For the results of the FPSB auction with 4 and 5 agents, see Fig 4 in the updated draft.

These results have been added to the paper and are reported in Appendix A.4.

Thank you once again to the reviewers for comments that inspired these robustness checks. We believe they improved the credibility of our results.

---

### Author Response · Authors · 2024-11-28
**Improving play from LLMs via learning**

A second theme in the reviewer responses were questions on learning. Namely, 1) When is there evidence of learning?, 2) what does it take to obtain learning behavior?

To answer both, we added a new Appendix section to the paper: Appendix A.7 (Learning).

Here, we repeat the results of the SPSB experiments in Section 3.1 but remove the ability for agents to conduct any chain of thought reasoning – we call such agents ‘Out-the-box’ agents.

Then, we compare the play of the 'Out-the-box' agents against 'Chain-of-thought' agents in the SPSB, and find:

1) Chain-of-thought agents perform better in the long run (after round 2, they consistently beat Out-the-box agents), and
2) Chain-of-thought agents demonstrate significant improvement in the first few rounds and then stabilize to a lower rate of mistakes, while Out-the-box agents demonstrate little to no improvement across 15 rounds of play.

We found it interesting that Chain-of-thought agents start with a much higher rate of mistakes in the first two rounds, suggesting that something about endowing agents with planning capacity causes them to 'tile the space of possible plans' more extensively when first trying out strategies.

Thank you again to the reviewers for the comments that inspired this additional work. We believe it greatly improved the paper.

Note: We made Appendix A.7 an extension of the setting in Section 3.1 (SPSB) because the SPSB auction has an equilibrium in dominant strategies of bidding one’s value. Hence, it's natural to plot the average absolute value deviation between bid and value (e.g., average |Bid - Value|) by round as a measure of ‘mistakes’ made by round -- if there is learning, we expected mistakes to go down by round (which they did for Chain-of-thought agents). Extensions could conduct a similar analysis by plotting mistakes to the theoretical benchmark under any equilibrium notion.

---

### Note · Authors · 2024-12-04

**Comment:**

We are very grateful to the reviewers for the close reading of our work and the numerous helpful comments.

Despite the very encouraging feedback, we must unfortunately withdraw the manuscript because we have been unable to complete the list of coauthors, which was left incomplete when submitting due to logistics, and thinking we could complete the author list after the deadline.

**Withdrawal Confirmation:**

I have read and agree with the venue's withdrawal policy on behalf of myself and my co-authors.